# Development and Validation of a Salt Food Frequency Questionnaire (FFQ-Na) and a Discretionary Salt Questionnaire (DSQ) for the Evaluation of Salt Intake among French-Canadian Population

**DOI:** 10.3390/nu13010105

**Published:** 2020-12-30

**Authors:** Maria Cecilia Gallani, Alexandra Proulx-Belhumeur, Natalie Almeras, Jean-Pierre Després, Michel Doré, Jean-François Giguère

**Affiliations:** 1Faculté des Sciences Infirmières, Université Laval, Québec, QC G1V 0A6, Canada; alexandra.proulx-belhumeur.1@ulaval.ca (A.P.-B.); michel.dore@fsi.ulaval.ca (M.D.); jean-francois.giguere@fsi.ulaval.ca (J.-F.G.); 2Centre de Recherche de l’Institut Universitaire de Cardiologie et de Pneumologie de Québec, Université Laval, Québec, QC G1V 4G5, Canada; Natalie.Almeras@criucpq.ulaval.ca; 3Département de Kinésiologie, Faculté de Médicine, Université Laval, Québec, QC G1V 0A6, Canada; jean-pierre.despres.ciussscn@ssss.gouv.qc.ca; 4VITAM—Centre de Recherche en Santé Durable, Québec, QC G1J 0A4, Canada

**Keywords:** Food-Frequency Questionnaire, discretionary salt, salt intake, measurement, reliability, validity

## Abstract

We assessed the reliability and validity of a Salty Food Frequency Questionnaire for Sodium (FFQ-Na) and a Discretionary Salt Questionnaire (DSQ) developed for the French-Canadian population. The reliability was evaluated according to temporal stability over a 7–15 day interval (*n* = 36). Validity was evaluated by testing the tools against a 24-h urine sodium excretion (24 h Uri-Na) and a 3-day food record, and this at individual and group levels (*n* = 164). The intra-class coefficients (ICC) values for the test-retest of the DSQ, the FFQ-Na and the two questionnaires combined were 0.73, 0.97 and 0.98 respectively. Correlations of the FFQ-Na with the 24 h Uri-Na and the 3-day food record were 0.3 (*p* < 0.001) and 0.35 (*p* < 0.001) respectively. The DSQ showed no significant correlation with the reference measures. The correlation between the two methods combined were 0.29 (*p* < 0.001) with the 24 h Uri-Na and 0.31 (*p* < 0.001) with the 3-day food record. The results of Bland–Altman indicated that for the combined questionnaires, there was a bias of measurement (underestimation of intake), but it was constant for every level of intake according to the reference measures. Finally, the cross-classification indicated an acceptable proportion of agreement, but a rate between 20% and 30% of classification in the opposite quartile. In conclusion, the developed tools are reliable and showed some facets of validity.

## 1. Introduction

There is a compelling body of evidence that high salt intake is an important and modifiable risk factor in the development and progression of high blood pressure [1,2], which is a significant risk factor for heart diseases, stroke and kidney failure [3,4]. High salt intake is also associated with premature death, dementia, disability, and increased healthcare costs worldwide [5,6]. Currently, the average salt intake is 10 g/day globally [7], when the recommended daily intake for healthy adults is <5 g/day (<2000 mg of sodium/day) [8]. Several intervention studies have shown that when populations reduce their salt intake, they show a reduction in their blood pressure [9,10]. A meta-analysis study has shown that even a modest daily reduction in salt intake for four weeks or more could lead to a significant decrease in blood pressure [11]. Therefore, there is a striking need for interventions aimed at promoting healthier salt intake levels [5], which requires first a close monitoring of salt intake as well as the identification of sources of salt in the diet [12,13].

The assessment of dietary salt intake, however, is not a straightforward process. Its overall intake results from complex nutritional behaviors and sources of intake are profoundly modulated by socioeconomic and cultural factors [14,15]. The 24-h urine sodium excretion (24 h Uri-Na), a recovery biomarker, is considered the gold standard method for the measurement of sodium intake [13,16], as approximately 85% to 90% of sodium ingested over a 24-h period is excreted in the urine. But this measure reflects only the recent intake (last 24 h) and it does not allow the identification of the sources of salt intake [15]. Thus, the use of self-reported questionnaires combined with a dietary biomarker is often recommended in the evaluation of salt consumption to improve the quality of the information, overcoming errors and better capture intra-individual variability of intake [17,18].

Self-reported measures of salt intake have often been used as exclusive measures to estimate salt intake in larger, longitudinal or epidemiological studies [19] since collecting 24-h urine samples in representative samples of the population can be challenging and resource-intensive [20]. Dietary 24-h recall, 3-day food record and weighed diet records are widely used as self-reported methods for the quantification of the nutrient intake. However, besides being labor intensive for both participants and researchers, they raise particular limitations regarding the assessment of salt intake: it is difficult to estimate precisely the sodium content in both processed and home-cooked food as well as the discretionary salt, the salt used during meal preparations or at the table [15].

Food-Frequency Questionnaires (FFQ)s have been widely used in nutritional epidemiology for the past 25 years [21]. FFQs can provide important information over longer periods of time, often a year. They also allow the identification of the major sources of salt intake, which is crucial for the elaboration of educational programs [22]. One of the limitations of the FFQs is that it is difficult to obtain a precise quantification of a nutrient. However, they assess intake over a longer period than dietary surveys, potentially overcoming problems associated with the high day-to-day variability of intake [15] and more importantly, they provide the main sources of salty foods that need to be targeted in educational programs. Discretionary salt can be a considerable source of total sodium intake and its assessment has been strongly recommended along with other self-reported methods [14,17]. Indeed, a recent review on the quality of FFQ specific for sodium intake recommends the assessment of discretionary salt along with the FFQ [15].

In Canada, the available information on salt intake is based on the National Nutrition and Health Survey, which used a 24-h food intake survey among adults and children [20] and to the best of our knowledge, there is no FFQ specifically for sodium intake (FFQ-Na) nor discretionary salt questionnaire (DSQ) validated for the Canadian population. The availability of such tools would allow to obtain a more detailed profile of salt intake among the population, thus, optimizing interventions aimed at promoting a healthier salt intake. Considering the specificities of cultural and environmental influences of the patterns of dietary intake, the aim of the present study was to develop and evaluate the measurement properties of reliability and validity of an FFQ-Na and a DSQ for the assessment of salt intake in the French-Canadian adult population. The hypotheses of the study were: (1) the FFQ-Na and the DSQ reliability results will demonstrate good temporal stability when the questionnaires are repeated within 7–15 days and (2) the FFQ-Na and the DSQ will provide evidence of validity according to individual and group criteria when compared to a 24 h Uri-Na and a 3-day food record.

## 2. Materials and Methods

### 2.1. Development of the Tools

The development of the FFQ-Na and the QSD was based on a previous experience with the Brazilian population [14] and on previously published studies [15,17,21]. Both instruments are available in Appendix A Texts S1 and S2.

#### 2.1.1. FFQ-Na

First, a list of high-Na foods frequently consumed by the French-Canadian population was compiled from previous surveys on eating habits among 26,856 Canadians of the target population [22]. The items with content ≥200 mg Na/100 g contributing potentially to the total sodium intake were selected [14,21,23]. Three experts in nutrition evaluated, for each food item, the total salt intake by portion, the relevance of the respective reference portion (tablespoon, cup, etc.) and the representativeness for the Canadian population [21,23]. The results of the three evaluations were pooled and a preliminary version of the FFQ-Na was re-evaluated by the experts. Upon a consensus, a 52-item version of the FFQ-Na was proposed in which participants indicate how frequently the food item was consumed during the last year on a scale ranging from 1 (never) to 7 (twice or more/day) as well as the portion usually consumed. We decided to use portion sizes in the questionnaire as recommended in the literature [17].

The estimation of the total sodium intake for each food item involved two steps: first, the frequency of intake was corrected for the frequency of monthly consumption (0 for the frequency score of 1, representing never; 0.5 for the frequency score of 2, representing less than once a month; 2 for a frequency score of 3, representing a frequency of 1–3 times a month; 4 for a frequency score of 4, representing once a week; 12 for a frequency score of 5, representing a frequency of 2–4 times a week; 30 for a frequency score of 6, representing a frequency of once daily; and 60 for a frequency score of 7, representing a frequency of twice or more daily), second, the adjusted frequency was multiplied by the nutritional value of salt for each item (previously identified by a nutritionist expert who used a food bank) according to the portion consumed (0.25, 0.5, 1, 2, etc.).

#### 2.1.2. DSQ

In the DSQ, participants were asked to rate the quantity of different types of salt used in the household. Several packages of the most commonly salt used by this population were presented to the participant (table salt, coarse salt, flower of salt, celery salt, garlic salt, seasoned salt and onion salt). Thus, for each of the salt indicated by the participant, he/she was asked to mention the duration of the package. Then, it was adjusted for the monthly use (example: a package of 500 g of table salt lasting for 6 months corresponded to the use of 83.3 g of table salt/month for the household). If the participant was not responsible for the grocery purchases and control of salt use at home, he/she was advised to ask for the social referent responsible for that. Afterwards all the types of salt used were summed and the amount of salt consumed per month was adjusted for the number of household members, who ate meals prepared at home, considering the number of meals per person and the age of the household members their age. Children under three years old were not considered in the calculation and for children between 3 and 10 years old, meals were considered as half-meals. This ultimately allowed us to estimate the discretionary salt consumption per person adjusted by daily use. For the calculation of sodium intake per person, the following steps were applied: (1) dividing the amount of salt (grams) used per month at home by 30 and multiplying by seven, (2) dividing this result by the total number of meals per week, providing the number of grams of salt consumed/meal, (3) multiplying the amount of salt/meal by the number of meals consumed by the participant, resulting in weekly salt intake and (4) dividing the participant’s total weekly salt intake by seven, resulting in an estimate of the individual daily amount of discretionary salt intake [14,20]. The same procedures were applied for salty seasonings. Visual cues as photos and empty packages of the mostly used types of salty seasonings were presented to assist participants when answering the questionnaire.

These first versions of the FFQ-Na and DSQ were pre-tested with a sample of 10 participants representing the sample of the validation step (men and women aged from 35 to 65 years living in Quebec City metropolitan area) invited from the surrounding community and accepting to participate as volunteers. A trained interviewer used cognitive interviewing techniques to assess the comprehensibility of the tools [24]. The results of this step reinforced the importance of using food models or simple shapes to represent home portion sizes as reference units [21].

### 2.2. FFQ-Na and DSQ—Reliability

The reliability of the proposed tools was evaluated according to the criterion of temporal stability, using the test-retest procedure over a 7–15-day interval, in a sample of 36 adults (19 males and 17 females) aged from 21 to 62 years (mean 45.9 ± 12.4 years).

### 2.3. FFQ-Na and DSQ—Validity

For the validation of data on salt intake provided by the FFQ-Na and the DSQ, both tools were tested against a 24 h Uri-Na (a biological biomarker) as well as a 3-day food record (a self-reported tool). For this step, 164 of the 332 asymptomatic adult men and women from the “Visceral obesity/ectopic fat and non-invasive markers of atherosclerosis: a cardiometabolic-cardiovascular imaging study (CMCV)” agreed to participate in this ancillary study. They were submitted to an additional visit, where they completed the questionnaires (FFQ-Na, DSQ and the 3-day food record). As previously proposed [17], all the questionnaires on salt intake were interviewer-administered by the trained research nurse or the research nutritionist, on a private room, using visual cues as realistic food models, photos, sample shapes and empty packages.

#### 2.3.1. 24 h Uri-Na

Participants also performed a 24 h urine collection as close as possible to the visit and no longer than over a 1-week interval. A 24 h urine collection was performed to measure sodium excretion, which allows to estimate the daily intake of dietary sodium. The participants were given the following instructions to carry out their urinary collect in an adequate way: (1) start the collection in the morning, eliminating the first urine of the morning, (2) collect all urinary volume in the 24-h period, including the first one of the next morning, (3) keep the container in the refrigerator throughout the collection until the scheduled return, and (4) keep usual eating and drinking habits during the period of collection. Urine was collected in polypropylene containers. When returning the urine collection, participants were asked about their adherence to the collection instructions provided. The total volume of the sample was measured and only volumes ≥500 mL were considered as a valid collection [15]. Urinary sodium was estimated in mEq/L by atomic absorption spectrometry of flame. The urinary sodium content in mEq was then converted in grams of salt, considering the molecular weight of sodium (1 mEq sodium = 23 mg of sodium = 0.058 g of salt) [5].

#### 2.3.2. 3-Day Food Record

Participants were informed by a nutritionist how to record all the food items they ate for 3 days, including a nonworking day. The 3-day food record was divided by meal (3 meals perday) and between meal (3 times per day) and included specifications regarding the location, time and quantity of each food consumed. All the food consumed by the participants were evaluated for daily energy intake, macronutrients, micronutrients such as sodium and fiber intake using Nutrific2001 (version 2.0; Department of Food Science and Nutrition at Laval, Laval University, Quebec City, Canada). A trained nutritionist reviewed the 3-day diary with each participant regarding the reported quantities and type of food. The 3-day average of intake was considered to quantify the salt consumed/day.

### 2.4. Anthropometry and Body Composition

According to standardized procedures, height, weight and waist circumference were measured [25,26]. Body mass index (BMI) was calculated. Body composition was assessed by Dual-Energy X-ray Absorptiometry (DEXA) using a Lunar Prodigy (GE Healthcare, Madison, WI, USA).

### 2.5. Hemodynamic Profile

Three sitting BP and pulse rate measurements were taken 3 min apart on the non-dominant arm with an appropriate cuff size. The mean values of the three measurements were retained [27]. BP was measured after the patient had been resting in the sitting position for at least 5 min. BP monitors did not change during the study.

### 2.6. Ethical Aspects

The local Institutional Review Board approved the study (2013-2127, 20908, 2012-1747, 20769) and all participants provided their written informed consent.

### 2.7. Statistical Analyses

All data obtained were doubly entered in a database created in the RedCap international database. SAS statistical software v9.2 (SAS Institute Biostatistic Inc., Cary, NC, USA) was used to analyze the data. Data were submitted to descriptive analyses and then to the evaluation of the measurement properties of reliability and validity. For the reliability assessment of the tools, the intra-class coefficients (ICC) were estimated to evaluate the proportion of agreement between the test and retest. The ICC was interpreted as values <0.5 representing poor reliability; values between 0.5 and 0.75, moderate reliability; values between 0.75 and 0.9, good reliability and values >0.9, excellent reliability [28]. For the validation of data on salt intake provided by the FFQ-Na and the DSQ, a combination of methods was used as recommended in the literature [15,29]. Both tools were tested against a 24 h Uri-Na as well as a 3-day food record according to the following analysis: (1) correlation coefficient (Spearman, as the distribution of the variables was not normal) to test the association between the tools at an individual level, (2) Bland-Altman to assess the agreement, presence and direction of bias at a group level and (3) cross-classification to evaluate the agreement at individual level (using quartile categories). The criteria proposed by Lombard were used for the interpretation of the analysis [29]. Thus, correlation coefficients >0.5 were considered as good outcomes; between 0.2 and 0.49 as acceptable outcomes and <0.2 as poor outcomes. For the Bland-Altman analysis, the presence, direction and extent of the bias were described and the absence of significant correlations between the differences and the means of the tools was considered as a good outcome. For the cross-classification, the criterion of an acceptable outcome was considered ≥50% of agreement in the same category and <10% in the opposite category. For this analysis, the LOGISTIC procedure was used. Values are presented as means and standard deviation as well as the median with the interquartile range (IQR). The extreme values (the mean +3 SD) were excluded for the inferential analyses. Finally, the size of our sample (*n* = 164) stands in the recommendations size [15,21,23].

## 3. Results

### 3.1. Reliability

The temporal stability of the tools was assessed according to the sodium intake provided for each item of the tools as well as their total score. The absolute value of the intake in the test retest for the overall score of the FFQ-Na and the DSQ as well as their respective ICC (95% CI) are presented in Table 1. ICC were respectively 0.73, 0.97 and 0.98 for DSQ, FFQ-Na and the combination of both, respectively, showing a good to excellent agreement (≥0.75).

For the DSQ sub-items, seasoning sauces, seasoning broths and seasoning spices, the ICC were respectively 0.85 (0.74–0.92), 0.78 (0.63–0.88) and 0.88 (0.78–0.93). For the items of the FFQ-Na, 5 of them presented an excellent agreement (>0.9), 13 good agreement (0.75–0.89), 22 moderate agreement (0.5–0.74) and 12 of them poor agreement (<0.5). It is important to mention that 10 out of the 12 items with poor agreement were consumed by less than 50% of the participants.

### 3.2. Validity

For the validation step, the 164 participants (male, 57.9%) were 54.1 (SD 8.0) years old with a BMI of 26.0 (SD 3.6) and an average blood pressure of 117/73 mmHg (SD 12/9). Descriptive characteristics are presented in Table 2.

The dietary intake of sodium from the DSQ, the FFQ-Na, the 3-day food record and the 24 h Uri-Na are summarized in Table 3. Dietary sodium intake as measured by 24 h Uri-Na, the standard reference, indicates an average consumption of 9.1 g of salt per day in the study population, exceeding the worldwide recommendations of <5 g/day [8], both for men and for women. In addition, men have a higher salt consumption than women according to the 24 h Uri-Na, as well as to the self-reported measures, with the exception of the DSQ which was higher for women than for men. Interestingly, the consumption of foods rich in salt (FFQ-Na) contributed more for sodium intake than the added salt (DSQ) for this population.

Correlation coefficients (Spearman) were estimated to test the association between the tools at individual level (Table 4). The FFQ-Na as well as the combination of the FFQ-Na and the DSQ presented acceptable correlation coefficients with the 3-day food record and the 24 h Uri-Na according to Lombard’s criteria [29], but the DSQ alone presented no significant correlations with the reference measures.

In order to assess the agreement, the presence and the direction of bias at a group level, Bland-Altman analysis was used (Figure 1a–f).

When the DSQ and FFQ-Na were considered separately with the reference measures, we observed a greater mean bias, and this difference was even more important for the DSQ that captures a smaller fraction of the salt intake in this sample. For the DSQ, mean bias with the reference measures were 6.9 g for the 24 h Na-Uri and 5.8 g for the 3-day food record. The FFQ-Na presented a mean bias of 4.2 g with the 24 h Na-Uri and of 3.3 g with the 3-day food record. When the measures were combined (FFQ-Na + QSD), the mean bias was 2.3 g with the 24 h Na-Uri and 1.4 g with the 3-day food record. Moreover, for the FFQ-Na and the DSQ separately, it was observed that the differences were more important as the mean intake was greater. That was confirmed by positive and significant correlations between the difference of measurements and the means. However, when the FFQ-Na and the DSQ were combined, the correlations were no longer significant, indicating that the differences with the reference measurements, although present, were constant (Table 5).

Finally, a cross-classification analysis was applied to evaluate the agreement at individual level, considering the quartiles of the measurements (Table 6). A 50% or more of agreement in the same category and no more than 10% in the opposite quintile are necessary to consider a good outcome [29]. We observed that for the FFQ-Na separately or in combination with the DSQ, the proportion of agreement was always about 50%, indicating a good level of agreement. However, the proportion of classification in the opposite quartile was more than 10% for all measures. This was also observed between the two reference measures, the 24 h Uri-Na and the 3-day food record.

## 4. Discussion

This study was aimed at presenting the measurement properties of two new tools developed to assess salt intake among the French-Canadian population: the FFQ-Na and the DSQ. These two measuring tools are important to: (1) the identification of the salty foods mostly consumed by the adult French-Canadian population, (2) the quantification of different types of salt usually added in the meals during or after their preparation and (3) the contribution of these sources to the overall salt intake.

The combination of both measurements is in line with the recommendations from the recent systematic literature review [30] on Food Frequency Questionnaires specific for sodium. The authors point the need to include in the FFQs an evaluation of the discretionary salt. The decision to develop a separate tool to assess the discretionary salt (DSQ) was based on the difficulties to estimate this intake in a unit basis because of the common use of the salt in the household. Consequently, it is difficult for the individual to estimate the precise amount of salt added to the meals.

The first hypothesis regarding the measurement properties of the FFQ-Na and the DSQ was related to the reliability, particularly the criterion of temporal stability. It was hypothesized that the results of the QSD and the FFQ-Na will be stable in the 7–15-day interval, considering that they are aimed to assess the salt intake on a 1-year period. We observed evidence of stability for the total scores of both tools. The ICC for the total score of the DSQ was of moderate magnitude, while for the total score of the FFQ-Na it was excellent [28]. For the combined tools, the coefficient remained excellent. The temporal stability of nutritional tools is an essential property to be evaluated because despite the day-to-day variability of intake, the foods listed by the participant must remain stable over the period covert by the tool. Our results seem to be among the best results reported in the literature for the FFQs in general that vary from 0.28 up to 0.9 [21]. These results support the evidence of temporal stability for the FFQ-Na and the DSQ.

Regarding the validity, our study examined different facets of this property, as recommended, at individual and group levels. Moreover, we tested the tools against two reference measurements, a 24 h Uri-Na and a 3-day food record. The 24 h Uri-Na was used since it is recognized as the gold standard in the quantification of salt intake, in spite of its well-known limits [13,15,16]. The 3-day food record was also used because as the FFQ-Na and the DSQ, it is a self-reported method, and it is considered an optimal comparison method since its sources of error are largely independent of error associated with dietary questionnaires. Additionally, it does not depend on memory, is open-ended and allows direct measurement of portion sizes [21].

Our first hypothesis regarding validity was that the tools would be positively and significantly correlated to the 24 h Uri-Na and the 3-day food record. Considering that the reference tools measure consumption over a smaller period and that the FFQ-Na and the DSQ only represents a portion of the salt intake, only correlations of moderated magnitude [29] were expected. The FFQ-Na presented correlation coefficients of 0.3 and 0.35 with the 24 h Uri-Na and the 3-day food record, respectively, which is aligned to the McLeans’ review [30] reporting correlations between the FFQs and the 24 h Uri-Na ranging from not statistically significant to moderate correlations (r ≤ 0.36). The DSQ alone, however, exhibited no correlations with the reference measures. To the best of our knowledge, there is a paucity of studies reporting the correlations between discretionary salt and a reference measure as the 24 h Uri-Na. In the McLeans’ review [30], only five out the 18 studies [14,31,32,33,34] evaluated the use of discretionary salt, and among these five studies, only one in a separate questionnaire [14]. It makes difficult to compare the performance of our DSQ to other similar tools.

The absence of correlation between the DSQ and the reference measures can be explained at least partiality by the fact that the individual salt addition estimated by the DSQ is derived from the total discretionary salt use at home. As consequence, in contrast to the measures of the FFQ-Na and the 3-day food record, which target specifically the individual intake, the DSQ does not capture the potential variation in the salt added to the meals by each member of the household. However, we recognize the limits of this tool in estimating the discretionary salt use. First, there is the limit of the recall bias associated with all self-reported methods in nutritional tools. Second, the calculation of the intake is based on an average for all the members of the household and it is not sensible for differences in the intake among them. With this in mind, it is important to highlight that precautions were taken to assure as much as possible the fidelity of the information: the questionnaire was administered rigorously by a nutritionist familiar with the questionnaire; visual cues of the packages of the different types of salt mostly used by the target population were provided; and the numbers of meals at home as well as the age of children were also considered. It is important to highlight that literature [15,17] reinforces the importance of considering the use of the DSQ, at least in the FFQ. Considering the particularities of the use of the discretionary salt, we decided to measure it by another tool. The contribution of the discretionary salt to the overall intake varies across populations; in the Brazilian study [14] it represented about 7 g of salt/day and in our population, it was about 2 g/day. Thus, is a source that in our population contributed to 40% of the maximum recommended value of consumption per day (<5 g/day) and should be evaluated. It is important to note that when both tools are added, the correlations with the reference measures are preserved. These findings support the evidence of validity concerning the association between the tools at individual level.

It was not expected that these two self-reported measures could represent the overall salt intake, since sodium is widely distributed in food, and the FFQ-Na and the DSQ are able to capture part of its intake. In fact, our data indicate that together, the FFA-Na and the DSQ were able to capture about 75% of the salt intake given by the 24 h Uri-Na, reinforcing the contribution of foods with high sodium content and of the discretionary salt in the overall salt intake in the targeted population. Alone, the FFQ-Na represented about 50% of the 24 h Uri-Na (4.7 out of 9.1 g/day), and the DSQ, about 22% (2 out of 9.1 g/day). The strong representation of the FFQ for the total salt intake can be explained by the fact that in Europe and North America, more than 75% of sodium intake is derived from processed and restaurant foods [35]. The salt added during or after preparation of meals contributes for only 15% of the overall intake [31].

The results of Bland-Altman test reinforce the fact that the FFQ-Na and the DSQ are partial measures of salt intake, and that even when combined, they tend to underestimate the total intake (2.3 g when compared to the 24 h Uri-Na and 1.4 g when compared to the 3-day food record). This tendency to underestimated intake is well known for FFQs [21]. However, when the tools are combined, in spite of the difference to the reference measures, this difference seems to be minimized and constant independently of the levels of sodium intake (r = 0.08; *p* = 0.3451 with the 24 h Uri-Na and r = 0.06; *p* = 0.4649 with the 3-day food record). These results indicate that for the combined measures, there is a bias of measurement, but it is constant for every level of intake. Furthermore, this bias is 70% (or 50%) less important when compared with another study reporting a difference of 5.5 g between their FFQ-Na and a 24 h Uri-Na [36]. Thus, our results support at least partially the hypothesis of agreement at group level.

Finally, concerning the cross-classification, we observed that the FFQ-Na separately or in combination with the DSQ was able to discriminate different levels of intake agreeing with the reference measures, and this for about 50% of the participants. The agreement for the DSQ was lower, just above 40%. However, there was a high proportion of classification on the opposite quartile: about 30% for the DSQ, 23 to 27% for the FFQ-Na and 27 to 29% for the combined tools. Thus, the tools combination neither contributed to optimizing the proportion of agreement neither to reduce the classification on the opposite quartile. These results indicate that our third hypothesis regarding the agreement at individual level was only partially confirmed, as the rate of opposite classification was higher than 10%, as recommended in the literature [29]. It is important to highlight, however, that the same was observed for the 3-day food record with the 24 h Uri-Na that presented a proportion of agreement higher than 50% but 22% of classification on the opposite quartile.

Several strengths must be highlighted in our study. First, the questionnaires assessed were developed for a specific population (French-Canadian adults), as strongly recommended in the literature [21]. Second, both tools allow identifying important sources of salt intake in this population, as the measurement of discretionary salt, often absent in the assessment of salt intake when using self-reported measures [15]. The estimation of the discretionary salt is not an easy task and there are several limits with its assessment. However, its measurement brings an important information about a household trend in the use of added salt. Third, two reference methods were used to test the hypotheses of validation of the tools—an objective (24 h Uri-Na) and a well-known self-reported measure method (3-day food record). Finally, we can highlight the use of rigorous standards in the development and evaluation of the self-reported tools for the assessment of salt intake from the conception of the tools to the robustness of the sample size, the use of interviewer-administered questionnaires and the variety of methods to assess their measurements properties.

As limits of our study, we can highlight the characteristics of our sample. The participants came from a larger study in which one of the inclusion criteria was to be asymptomatic adults. We can see this in the characterization of the sample, which had relatively low values of BMI, waist circumference and blood pressure, which is not in line with the profile of the overall population. Canada presents a rate of about 25% of obesity [37] and 17% of hypertension [38], all of them associated with poor dietary habits and high salt intake. Thus, further studies, including a more diversified population in terms of proportion and intensity of risk factors for cardiometabolic diseases, are recommended to continue the evaluation of the measurement properties of the developed tools. Another important limit of our study was the use of only one measure of 24 h Uri-Na. First, our tools and the reference measures did not assess salt intake in the same reference period. Second, 24 h Uri-Na, although recognized as the gold standard for estimating salt intake, is also associated with significant intra-individual variability under controlled environment [39] which further contribute to the limitation of our findings. Thus, the inclusion of several measures of 24 h Uri-Na (recommendations are from two to ten [15]) will be important to be planned in further studies. Finally, concerning the 24 h urine collection, although participants were asked about their adherence to the collection and that only total volume ≥ 500 mL were considered as a valid collection, no other methods such as urinary creatinine-based strategies or *p*-aminobenzoic acid (PABA) excretion were used to assess completeness of urinary collection.

## 5. Conclusions

In conclusion, our data indicate that, when applied to a sample of asymptomatic French-Canadian adults, the FFQ-Na and the DSQ demonstrated evidence of reliability and partial validity at individual and group levels. Further studies are recommended to carry on the assessment of the validity of these tools in a more diversified population.

The results of this study are important for the French-Canadian population. First, because we found a total intake of about 9 g of salt/day according to the 24 h Uri-Na, surpassing the WHO recommendations of 5 g to 6 g/day [8]. This finding reinforce the importance of addressing the problem of salt intake in this population. Second, because we provide two self-reported measures of salt intake, which along with the objective measures of 24 h Uri-Na can subside further studies to understand the impact of salt intake on health as well as to inform public health workers for dietary sodium reduction, considering the specificities of dietary patterns of this population [30].

## Figures and Tables

**Figure 1 nutrients-13-00105-f001:**
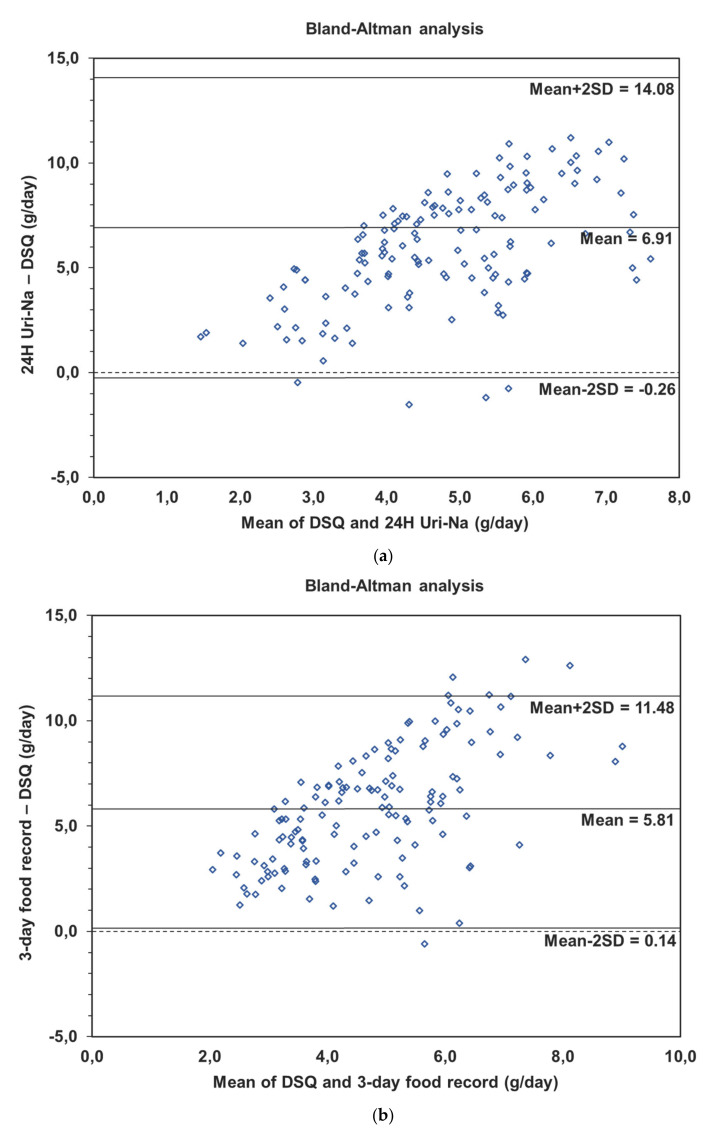
(**a**) Bland-Altman for the DSQ and the 24 h Uri-Na. (**b**) Bland-Altman for the DSQ and the 3-day food record. (**c**) Bland-Altman for the FFQ-Na and the 24 h Uri-Na. (**d**) Bland-Altman for the FFQ-Na and the 3-day food record. (**e**) Bland-Altman for the FFQ-Na + DSQ and the 24 h Uri-Na. (**f**) Bland-Altman for the FFQ-Na + DSQ and the 3-day food record. DSQ: Discretionary Salt Questionnaire; FFQ-Na: Salt Food Frequency Questionnaire; FFQ-Na + DSQ: Salt Food Frequency Questionnaire + Discretionary Salt Questionnaire; 24 h Uri-Na: 24 h urinary sodium excretion.

**Table 1 nutrients-13-00105-t001:** Total salt intake assessed from the test-retest for the DSQ and the FFQ-Na (*n* = 36).

	Test	Retest	ICC (95% CI)
DSQ (g/day)	1.7 ± 1.2	1.5 ± 1.0	0.73 (0.56, 0.85)
FFQ-Na (g/day)	4.7 ± 2.3	4.6 ± 2.2	0.97 (0.94, 0.98)
FFQ-Na+ DSQ (g/day)	6.4 ± 2.7	6.3 ± 2.7	0.98 (0.95, 0.99)

DSQ: Discretionary Salt Questionnaire; FFQ-Na: Salt Food Frequency Questionnaire; FFQ-Na+ DSQ: Salt Food Frequency Questionnaire + Discretionary Salt Questionnaire; ICC: Intra-Class Coefficient; CI: Confidence interval.

**Table 2 nutrients-13-00105-t002:** Sociodemographic and clinical characteristics of the participants who participated in the validity substudy.

	Total Sample (*n* = 164)		Male (*n* = 95)		Female (*n* = 69)	
	Mean (SD)	Median (IQR)	Mean (SD)	Median (IQR)	Mean (SD)	Median (IQR)
Age (years)	51.4 (8.0)	52.1 (14.2)	51.1 (7.9)	51.6 (13.5)	51.9 (8.1)	53.3 (14.6)
BMI (kg/m^2^)	26.0 (3.6)	25.8 (4.6)	26.3 (2.9)	26.3 (3.6)	25.5 (4.4)	24.6 (5.8)
WC (cm)	89.5 (11.1)	90.3 (16.9)	93.5 (9.3)	92.7 (11.2)	83.7 (11.0)	82.4 (18.4)
SBP (mmHg)	117 (12)	118 (8)	120 (10)	120 (15)	113 (12)	114 (18)
DBP (mmHg)	73 (9)	72 (12)	75 (8)	75 (12)	70 (9)	68 (13)

WC: waist circumference, SBP: systolic blood pressure, DBP: diastolic blood pressure, SD: standard deviation, IQR: interquartile range.

**Table 3 nutrients-13-00105-t003:** Dietary intake of sodium from the DSQ, the FFQ-Na, the 3-day food record and the 24 h Uri-Na.

	Total Sample		Male		Female		*p* Value
	Mean(SD)	Median(IQR)	Mean(SD)	Median(IQR)	Mean(SD)	Median(IQR)	
DSQ *(*n* = 164)	2.0(1.6)	1.6(1.7)	1.8(1.4)	1.4(1.1)	2.4(1.9)	1.8(1.8)	*p* = 0.022
FFQ-Na *(*n* = 164)	4.7(2.3)	4.3(2.6)	5.6(2.4)	5.1(2.5)	3.6(1.7)	3.3(1.6)	*p* < 0.0001
3-day food record *(*n* = 142)	7.9(2.8)	7.7(1.7)	8.9(2.8)	8.8(2.0)	6.6(2.2)	6.4(1.4)	*p* < 0.0001
24 h Uri-Na *(*n* = 162)	9.1(4.2)	8.2(4.5)	10.3(4.5)	9.4(4.6)	7.6(3.2)	7.1(3.6)	*p* < 0.0001

* g of salt/day. SD: standard deviation; IQR: inter-quartile range.

**Table 4 nutrients-13-00105-t004:** Spearman correlation coefficients between the DSQ and the FFQ-Na, as well as the correlations of the DSQ, the FFQ-Na and the FFQ-Na + DSQ, with the 3-day food record and 24 h Uri-Na.

	DSQ	FFQ-Na	FFQ-Na + DSQ
FFQ	0.13		
3-day food record	0.03	0.35 *	0.31 *
24 h Uri-Na	0.09	0.30 *	0.29 *

* *p* < 0.001. DSQ: Discretionary Salt Questionnaire; FFQ-Na: Salt Frequency Food Questionnaire; FFQ-Na + DSQ: Salt Frequency Food Questionnaire + Discretionary Salt Questionnaire.

**Table 5 nutrients-13-00105-t005:** Bland-Altman results for the DSQ, FFQ-Na, FFQ-Na + DSQ, 3-day food record and 24 h Uri-Na.

	Difference (SD); (mean − 2SD, mean + 2SD)	Correlation
DSQ and 3-day food record (*n* = 136)	5.8 (2.8); (0.1, 11.5)	0.56 *p* < 0.0001
DSQ and 24 h Uri-Na (*n* = 141)	6.9 (3.6); (−0.3, 14.1)	0.61 *p* < 0.0001
FFQ-Na and 3-day food record (*n* = 138)	3.3 (2.6); (−1.9, 8.5)	0.38 *p* < 0.0001
FFQ-Na and 24 h Uri-Na (*n* = 144)	4.2 (3.5); (−2.9, 11.2)	0.28 *p* = 0.0006
FFQ-Na + DSQ and 3-day food record (*n* = 137)	1.4 (2.9); (−4.4, 7.3)	0.06 *p* = 0.4649
FFQ-Na + DSQ and 24 h Uri-Na (*n* = 144)	2.3 (3.8); (−5.2, 9.8)	0.08 *p* = 0.3451

SD: standard deviation; DSQ: Discretionary Salt Questionnaire; FFQ-Na: Salt Food Frequency Questionnaire; FFQ-Na + DSQ: Salt Food Frequency Questionnaire + Discretionary Salt Questionnaire; 24 h Uri-Na: 24 h urinary sodium excretion.

**Table 6 nutrients-13-00105-t006:** Cross-classification results for the DSQ, FFQ-Na, FFQ-Na + DSQ, 24 h Uri-Na and 3-day food record.

	Proportion of Agreement	Proportion of Opposite Quartile
DSQ and 24 h Uri-Na	41.4%	33.8%
DSQ and 3-day food record	43.8%	32.3%
FFQ-Na and 24 h Uri-Na	53.3%	23.1%
FFQ-Na and 3-day food record	48.3%	27.6%
FFQ-Na + DSQ and 24 h Uri-Na	49.2%	27.1%
FFQ-Na + DSQ and 3-day food record	48.0%	27.8%
24 h Uri-Na and 3-day food record	54.3%	22.0%

DSQ: Discretionary Salt Questionnaire; FFQ-Na: Salt Food Frequency Questionnaire; FFQ-Na + DSQ: Salt Food Frequency Questionnaire + Discretionary Salt Questionnaire; 24 h Uri-Na: 24 h urinary sodium excretion.

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
