# Peer review of "Development and Validation of a Salt Food Frequency Questionnaire (FFQ-Na) and a Discretionary Salt Questionnaire (DSQ) for the Evaluation of Salt Intake among French-Canadian Population"

_nutrients, 2020, doi:10.3390/nu13010105_

Round 1

Reviewer 1 Report

 Gallani et al. assessed the reliability and validity of a Salty Food Frequency Questionnaire for Sodium (FFQ-Na) and a Discretionary Salt Questionnaire (DSQ) developed for the French-Canadian population. The reliability was evaluated over a 7-15 day interval in 36 subjects whereas validity was assessed by testing the tools against a 24-hour urine sodium excretion (24h Uri-Na) and a 3-day food record in 164 individuals. The authors found intra-class coefficients (ICC) values for the test-retest of the DSQ, the FFQ-Na and the two questionnaires combined of 0.73, 0.97 and 0.98 respectively. Spearman correlations coefficients of the FFQ-Na with the 24h Uri-Na and the 3-day food record were 0.30 and 0.35 respectively. The DSQ showed no significant correlation with the reference measures. The correlation between the two methods combined were 0.29 with the 24h Uri-Na and 0.31 with the 3-day food record. Bland–Altman results indicated that for the combined questionnaires there was a systematic underestimation of intake, albeit constant for every level of intake. The cross-classification indicated an acceptable proportion of agreement but a high rate of misclassification in the opposite quartile. The authors concluded that the tools under investigation are reliable and show partial validity.

This is an interesting piece of work on an important issue, i.e. the investigation of suitable methods for the assessment of group and individual sodium intake in population studies. Indeed, other works have already explored similar pathways with generally disappointing results. The instruments set up by the authors for the French Canadian population undoubtably show some merit although overall they confirm the difficulty to build up a tool reliable and valid enough to replace 24h urine collection for the assessment of salt intake at group level , let alone the assessment of individual consumption.

Specific points:

  • Materials and Methods 2.1 The authors state that “the development of the FFQ-Na and the QSD was based on a previous experience with the 89 Brazilian population [14] and on previously published studies [15, 17, 21].” Do they mean that the questionnaire contents are already published? If not, are they willing to add an appendix to the article displaying the questionnaires items?

  • Materials and Methods 2.1.2. “In the DSQ, participants were asked to rate the quantity of grams of salt used per month for the preparation of meals and for the salt added at the table for the household”. I really wonder how the authors obtained this type of information: I think almost no one is in the position to know the monthly amount of salt used in the kitchen and table by the family. May be some assumption may be made by the person who prepares foods for the family but, unless you are prepared to this type of request, it is difficult any way. I would exclude that any other member of the family is in the position to answer this question with a reasonable degree of approximation. Could the authors expand on this point with some more convincing explanation?

  • The authors seem to be relatively satisfied with the correlation found between the FFQ-Na and the 24h Uri-Na / 3-day food record (0.30 and 0.35, respectively): however, these values actually indicate that only 9 - 12% of the variation in sodium intake (as estimated by the two reference tools) was caught by their FFQ-Na, not a great result unfortunately.

  • The authors recognize the use of a single 24h urine collection as a limitation: this is correct but I think they should recognize that this is a very important limitation since they are looking at correlations at the individual level between the result of 24h urine collection and the tools they are trying to validate: there is no way to use a single collection to estimate sodium intake at the individual level. In this respect, it is not surprising that the FFQ-Na result was better related to the 3-day food record than to the 24h urine collection results.

Author Response

RESPONSE TO REVIEWERS COMMENTS

Manuscript #:              Nutrients-1031853.R1

Authors:                      Gallani, Maria-Cecilia et al.

Title:                           Development and validation of a Salty Food Frequency Questionnaire for Sodium (FFQ-Na) and a Discretionary Salt Questionnaire (DSQ) for the evaluation of salt intake among French-Canadian

We thank the reviewer for their valuable and constructive suggestions and comments on the paper, which has undoubtedly, improved our manuscript. We hope that this detailed point-by-point rebuttal will satisfactorily address all comments/issues. The modifications in the paper were conducted in the revision mode.

Best regards,

Maria Cecilia Gallani, on behalf of all co-authors.

Specific points:

1- Materials and Methods 2.1 The authors state that “the development of the FFQ-Na and the QSD was based on a previous experience with the 89 Brazilian population [14] and on previously published studies [15, 17, 21].” Do they mean that the questionnaire contents are already published? If not, are they willing to add an appendix to the article displaying the questionnaires items?

Reply: Thank you for the comment. In fact, the contents were not published yet. Thus, as requested, both instruments (FFQ-Na and DSQ) are now available in Appendix 1.

2- Materials and Methods 2.1.2. “In the DSQ, participants were asked to rate the quantity of grams of salt used per month for the preparation of meals and for the salt added at the table for the household”. I really wonder how the authors obtained this type of information: I think almost no one is in the position to know the monthly amount of salt used in the kitchen and table by the family. May be some assumption may be made by the person who prepares foods for the family but, unless you are prepared to this type of request, it is difficult any way. I would exclude that any other member of the family is in the position to answer this question with a reasonable degree of approximation. Could the authors expand on this point with some more convincing explanation?

Reply: Thank you very much for this remark. In fact the description of the steps taken to obtain the information of the discretionary salt were oversimplified in the text. Thus, we rephrased as follows:

In the DSQ, participants were asked to rate the quantity of different types of salt used in the household. Several packages of the most commonly salt used by this population were presented to the participant (table salt, coarse salt, flower of salt, celery salt, garlic salt, seasoned salt and onion salt). Thus, for each of the salt indicated by the participant, he/she was asked to mention the duration of the package. Then, it was adjusted for the monthly use (example: a package of 500 g of table salt lasting for 6 months corresponded to the use of 83,3 g of table salt/month for the household). If the participant was not responsible for the grocery purchases and control of salt use at home, he/she was advised to ask for the social referent responsible for that. Afterwards all the types of salt used were summed and the amount of salt consumed per month was adjusted for the number of household members, who ate meals prepared at home, considering the number of meals per person and the age of the household members their age. Children under three years old were not considered in the calculation and for children between 3 and 10 years old, meals were considered as half-meals.  This ultimately allowed us to estimate the discretionary salt consumption per person adjusted by daily use.

And this following sentence was added in the discussion section:

However, we recognize the limits of this tool in estimating the discretionary salt use. First, there is the limit of the recall bias associated with all self-reported methods in nutritional tools. Second, the calculation of the intake is based on an average for all the members of the household and it is not sensible for differences in the intake among them. With this in mind, it is important to highlight that precautions were taken to assure as much as possible the fidelity of the information: the questionnaire was administered rigorously by a nutritionist familiar with the questionnaire; visual cues of the packages of the different types of salt mostly used by the target population were provided, and the numbers of meals at home as well as the age of children were also considered.

It is important to highlight that literature [15,17] reinforces the importance of consider the use of the DSQ, at least in the FFQ. Considering the particularities of the use of the discretionary salt, we decided to measure it by another tool. The contribution of the discretionary salt to the overall intake varies across populations; in the Brazilian study [14] it represented about 7.0 g of salt/day and in our population it was about 2.0 g /day.  Thus, is a source that in our population contributed to 40% of the maximum recommended value of consumption per day (<5g/day) and should be evaluated.

3- The authors seem to be relatively satisfied with the correlation found between the FFQ-Na and the 24h Uri-Na / 3-day food record (0.30 and 0.35, respectively): however, these values actually indicate that only 9 - 12% of the variation in sodium intake (as estimated by the two reference tools) was caught by their FFQ-Na, not a great result unfortunately.

Reply: In fact, its not a huge correlation, but when we look to other similar papers on self-reported  tools and more specifically on FFQ-Na, it is the same pattern observed, as pointed in the discussion section (Line 330): McLeans’ review (2014) reporting correlations between the FFQs and the 24h Uri-Na ranging from not statistically significant to moderate correlations (r≤0.36)”.   It is important to note that this weaker correlation in the FFQ-Na and 24h Uri-Na is explained also by the fact that Na is ubiquitously present in several types of foods. It is different from other nutriments as carotene for which is possible to target the more important sources of the nutrient. Thus, the correlation between self-reported and biological measures is expected to be much higher.

4- The authors recognize the use of a single 24h urine collection as a limitation: this is correct but I think they should recognize that this is a very important limitation since they are looking at correlations at the individual level between the result of 24h urine collection and the tools they are trying to validate: there is no way to use a single collection to estimate sodium intake at the individual level. In this respect, it is not surprising that the FFQ-Na result was better related to the 3-day food record than to the 24h urine collection results.

Reply: We agree with the reviewer regarding the limits of using a single 24h urine collection for assessing correlations at individual level. Thus, we added a clarification about this limit in the last paragraph of the discussion (line 416)

“Finally, concerning the 24h urine collection, although participants were asked about their adherence to the collection and that only total volume ≥ 500 ml were considered as a valid collection, no other methods such as urinary creatinine-based strategies or p-aminobenzoic acid (PABA) excretion were used to assess completeness of urinary collection”.

Reviewer 2 Report

This is a well written and designed study.  

1. The completeness of the 24hr urine collections are important to measuring sodium intake. Assessment of the completeness of the 24hr urine collections can be strengthened using the Creatinine Index >0.7 for under collections and over collections assessed by 24hr urine creatinine > 20mg/kg/day for females and > 25mg/kg/day for males.  If creatinine was measured, suggest including this assessment for the robustness of the analysis, otherwise include comment in limitations. 

2. For the 3-day food record, were there any instructions regarding which days of the week the participants would capture? Where any of the days weekends included?

3. Line 142: "For this step, 164 on the 332 asymptomatic...." should be "...164 of the 332.."

4. Line 206: "(IQR)The extreme values..." should be "(IQR). The extreme values..."

Author Response

RESPONSE TO REVIEWERS COMMENTS

Manuscript #:              Nutrients-1031853.R1

Authors:                      Gallani, Maria-Cecilia et al.

Title:                           Development and validation of a Salty Food Frequency Questionnaire for Sodium (FFQ-Na) and a Discretionary Salt Questionnaire (DSQ) for the evaluation of salt intake among French-Canadian

We thank the reviewer for their valuable and constructive suggestions and comments on the paper, which has undoubtedly, improved our manuscript. We hope that this detailed point-by-point rebuttal will satisfactorily address all comments/issues. The modifications in the paper were conducted in the revision mode.

Best regards,

Maria Cecilia Gallani, on behalf of all co-authors.

Comments and Suggestions for Authors

This is a well written and designed study.  

  1. The completeness of the 24hr urine collections are important to measuring sodium intake. Assessment of the completeness of the 24hr urine collections can be strengthened using the Creatinine Index >0.7 for under collections and over collections assessed by 24hr urine creatinine > 20mg/kg/day for females and > 25mg/kg/day for males.  If creatinine was measured, suggest including this assessment for the robustness of the analysis, otherwise include comment in limitations. 

Reply: Thank you for your comments. Unfortunately, we did not consider those methods to assess completeness of urine collection in our study. Such limitation has now been briefly discussed in the last paragraph of the discussion (line 416) :

“Finally, concerning the 24h urine collection, although participants were asked about their adherence to the collection and that only total volume ≥ 500 ml were considered as a valid collection, no other methods such as urinary creatinine-based strategies or p-aminobenzoic acid (PABA) excretion were used to assess completeness of urinary collection”.

  1. For the 3-day food record, were there any instructions regarding which days of the week the participants would capture? Where any of the days weekends included?

Reply:  Information about the 3-day dietary journal have been added to the method section as follow:

Participants were informed by a nutritionist how to record all the food items they ate for 3 days, including a nonworking day.

  1. Line 142: "For this step, 164 on the 332 asymptomatic...." should be "...164 of the 332.."

Reply: Thank you to the reviewer to revised the text.

  1. Line 206: "(IQR)The extreme values..." should be "(IQR). The extreme values..."

Reply: Thank you to the reviewer to revised the text.

Round 2

Reviewer 1 Report

The authors have done their best to meet this reviewer's comments and suggestions.